# Proximity Indexing of Public Transport Terminals in Metro Manila

**Krister Ian Daniel Roquel \***, **Raymund Paolo Abad**  **and Alexis Fillone**

Civil Engineering Department, De La Salle University, Manila 1004, Philippines;
raymund.abad@dlsu.edu.ph (R.P.A.); alexis.fillone@dlsu.edu.ph (A.F.)
**\*** Correspondence: krister.roquel@dlsu.edu.ph

**Abstract:** Despite the extensive transit network in Metro Manila, intermodal connections between public transportation services are still fragmented. In response, authorities proposed various multimodal transport terminals around the periphery of the metropolis. However, there is a need to understand how these proposed terminals will impact existing transportation infrastructure and services as well as the current travel demand. This paper proposes a method that quantifies the nearness of any subject to any metric of interest, or in this case, the location of the terminal based on its proximity to existing transit supply and demand at different points in the transport network. It involves a simple methodology that requires only the spatial distribution of relevant transport planning data (e.g., public transport services, public transport passenger activity). It was found that the spatial distribution of the transport terminals in the study area is more closely related to the transit supply. Using the same methodology, several potential locations in Metro Manila (e.g., central terminal, terminal along a major junction) were assessed to see whether these are viable sites for a multimodal terminal. One scenario configuration was found to be better integrated with where trips start and/or end, while another seemed to improve integration of the existing railways.

**Keywords:** intermodal; transit demand; transit supply; Metro Manila

## 1. Introduction

An intermodal trip consists of more than one stage of transport, [1] using at least two different transport modes, lines, or operators [1–6] within a single trip or path [7]. Intermodal trips occur due to public transport's inability to support door-to-door services using a singular travel mode [8]. Hence, it comes as no surprise that transferring between different transport modes in large public transport networks is inevitable, [9,10] such as in major cities including London, New York, Munich, Paris, and Melbourne [9,11]. Studies estimate that mobility in the future will be increasingly intermodal [12] as different travel modes become increasingly available [13] and mobility patterns become more individualized [14] and adaptive to meet personal needs [7,15–17]. Despite this, transferring between travel modes remains inconvenient [7,9] and costly even for developing countries [18], because of the uncertainties [19] associated with the lack of integration and cooperation between travel modes. Individuals may have a negative perception because of this inconvenience [20], and it may discourage them from considering public transportation as a potential alternative to private car travel [21–23].

Because intermodal transfers are more tedious to navigate than intramodal transfers [24], transport interchanges or terminals are critical nodes in the transport network [25], responsible for ensuring fast and minimal effort in changing between at least two modes of transportation [11,26]. These facilities are considered more complex than conventional stations [27] because it also provides the physical integration of various means of transport [28] such as local or regional transport [29]. These complexities require planners to carefully evaluate the location and accessibility of these facilities to maximize the benefits [11,12] of intermodal travel, such as enhancing public transport use and reducing

car dependence [24,30–32], lowering $CO_2$ emissions when controlling travel distance [4], increasing passenger satisfaction [5], and promoting sustainable development [6]. Overall, users of the transport system will benefit from an integrated transport system because of the improvements in connectivity and convenience it brings. Developed coordination between various transport modes also improves economic productivity and efficiency, as well as the mobility of vulnerable populations, including the elderly, disabled, and economically disadvantaged. Aside from the social benefits, an efficient multimodal transportation system will also have positive effects on the environment. As such, careful planning of the transportation system makes for the sustainable and efficient use of finite resources.

The location of transport terminals is crucial to their performance, quality, and importance [11,24,33]. In some instances, the similarities and differences between terminals typically depend on the context of their location [11]. Arora and Chanda [33] listed several factors that could aid planners to identify the location of a terminal. In their report, factors such as regional connectivity, existing ridership, route convergence, and current road network configuration should be considered. For Rodrigue et al. [34], terminals should be in an area in which it can serve large economic activities. These varying factors taken into consideration led to different locations of terminals depending on their function. For instance, Bernal [26] found that the intermodal facilities in eight cities were mostly located at nodes around central business districts and in areas where private vehicles are competitive with public transport. In Monzon et al. [12], interchanges were located at either the center or the outskirts of the city to cater to long-distance, last-mile connections. Finally, some old railway stations are being converted into intermodal public transport hubs because these are located at the center of cities and towns [35].

There are different studies that either evaluated or determined the location of an intermodal transport terminal. In the logistics field, studies addressed the terminal location problem using various techniques. For instance, locations of intermodal transport terminals were determined using spatial analysis with cost and environmental goals [36]. The technique evaluated alternative locations in the transport system that produce the least environmental impact without compromising the quality of service. Several works focused on evaluating essential criteria to locate intermodal terminals. The analytic hierarchy process (AHP) revealed that decision-makers allocated the most weight to efficient flows among other criteria in selecting the terminal location in Croatia [37]. A hybrid Strengths-Weaknesses-Opportunities-Threats (SWOT) / Multiple-Criteria Decision-Making (MCDM) model determined the location of a rail–road intermodal terminal [38]. The researchers first conducted a SWOT analysis of the proposed terminals before determining the criteria and sub-criteria's weights using AHP. Meanwhile, a study [39] addressed the ambiguity and uncertainty of the evaluation process of decision-makers by introducing fuzzy methods in their model. Programming methods [40,41] and mathematical models [42] also provided solutions to the terminal location problems to minimize costs.

Although studies are limited [43], researchers applied similar approaches in the planning and locating of transit terminals. Spatial analysis revealed the clustered passenger terminals in Lagos [44] and the relationship of the distribution of bus terminals in Wuhan, China [45]. The work in China was extended by locating candidate bus terminals by considering the spatial distance, road density, and forecasted passenger demand and by acquiring additional terminals through a buffer approach. Studies also employed multicriteria analyses in evaluating transit interchanges [29], bus terminals [46], and transit hubs [47], using various criteria such as spatial conditions, accessibility to other transport modes, and basic and specific interchange elements. Methods that use multi-criteria analysis weigh these criteria according to expert or stakeholder opinions that would enable them to evaluate or compare terminal locations. Programming methods also aided in the location of transit network elements. Researchers developed a nonlinear program that locates transfer terminals by planning the public transport network and then adjusting the terminal locations until equilibrium objectives are met [43]. Other studies optimized the location of a bus depot [48] and bus stops [49]. The programs identified the locations by

minimizing the operational costs of assigning buses to depots [48] and the economic costs of transfers [49].

The authors of this study present an alternative method using an index measure for evaluating the location of an intermodal terminal based on its immediacy to both transit supply and demand. Unlike the methods in the previously mentioned studies, this method explicitly evaluates the nearness of the proposed terminal to existing routes and passenger movements at different nodes in the transport network. The index could indicate whether the terminal is nearer to transport supply or to travel demand. The results could be used by planners and decision-makers to maximize benefits for transport users and operators using the terminal.

This study is conducted in a developing country where there is still little research on coordinating multimodal public transit [18]. The public transit network of Metro Manila is presented as a case study because the transit network is marred with integration issues, despite the high demand for transit services [50] and evidence of intermodal travel [51]. Local authorities are investing in intermodal interchanges located in and around Metro Manila not only to support public transit infrastructure but also to curb worsening traffic congestion [52] by limiting the number of operating non-rail-based transit services, such as buses and jeepneys in the metropolis. The proposed indices evaluated existing terminal locations with respect to current transit supply and demand configurations as well as the proposed locations of additional transit terminals.

The next section briefly discusses the Metro Manila transportation network. Section 3 describes the framework of the indices. Section 4 discusses the spatial distribution of transit supply and demand. Section 5 provides an analysis of the additional transit terminals. Section 6 concludes the study.

## 2. Case Study: The Metro Manila Transportation Network

The public transportation network of Metro Manila (shown in Figure 1) consists of rail lines, buses, jeepneys, Asian utility vehicles (AUVs), and three-wheeled motor- or human-powered tricycles or pedicabs, respectively. Public transit services carry the bulk of travel (69.6%) [53,54] because of an extensive road-based transit network (shown in Figure 2). Table 1 summarizes the service characteristics of the different public transit services. Jeepneys, although designed mostly for intra-city services along primary and secondary roads, dominate the network coverage of about 3.5 million route kilometers and roughly 35,000 franchised units [55]. Buses are second to jeepneys when it comes to network coverage.

**Table 1.** Service operating characteristics of public transport services in Metro Manila.

| Mode | Jeepney | Bus | AUV | Rail |
|---|---|---|---|---|
| No. of routes | 216 | 151 | 109 | 4 |
| Ave. Length (km) | 8.62 | 31.37 | 17.63 | 29.08 |
| Ave. Speed (kph) | 15.06 | 20.02 | 17.72 | 33.75 |
| Seated Capacity | 20 | 40 | 10 | 232 |
| Total Capacity | 20 | 60 | 10 | 1628 |
| Basic Fare (PHP), 1 USD = 48.95 PHP | 9 | 12 | 15 | 15 |

In 2012, there were about 2250 bus units, less than 10% of the number of jeepneys. Because of its larger seating capacity and longer service routes ranging from 4.2 to 60 km, buses account for more than half of the daily commute trips. However, bus services only account for about 14.4% of the total modal share among public modes of transport, second to jeepneys (36.3%) and ahead of rail modes (11.2%). Last among the road-based transit modes are AUVs (including FX or UV Express), which function as shared taxis. Route ends are given to their franchises, which means drivers are free to take any roads to reach their destination. To avoid competition with jeepneys, AUVs operate on a point-to-point basis, disallowing loading and unloading along its route. Despite AUVs having the least route

coverage compared to jeepneys and buses, it has been widely patronized by employees working in central business districts (CBDs) because it is a faster alternative to other road-based transit modes [55]. Analysis of the trips taken per passenger revealed that a passenger's typical trip consists of 1.56 legs [51], indicating that the trips in the region may be intermodal.

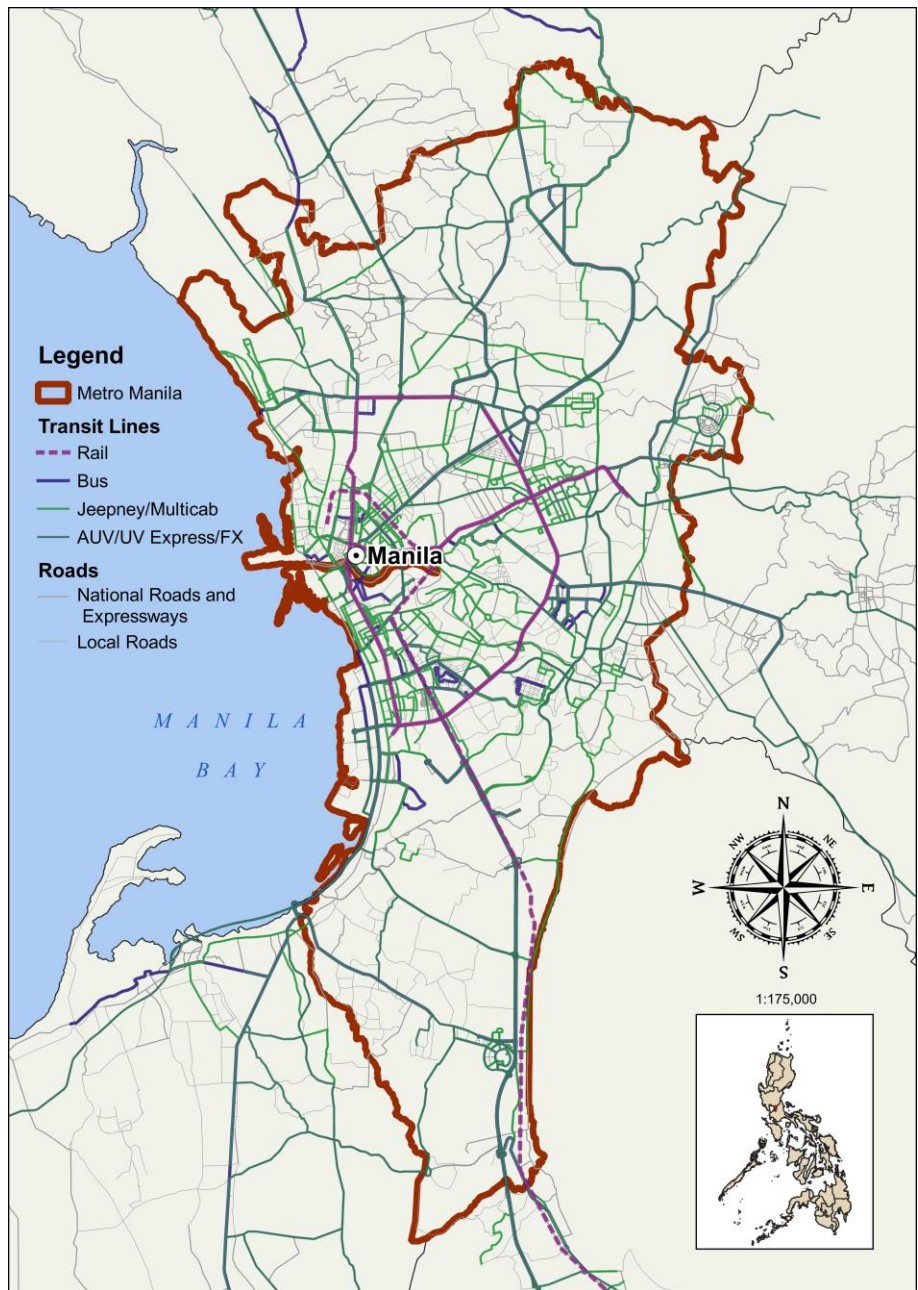

**Figure 1.** Metro Manila road transport infrastructure and public transport services.

Despite the extensive network in Metro Manila, intermodal connections between public transportation services are still fragmented. Specifically, there is an absence of functional connections between mass transit services. For instance, the northern end of MRT 3 and LRT 1, [51] and LRT 1 and LRT 2, are characterized by long walks between lines. In other stations where transfers of passengers between transit modes occur, facilities are improperly designed to accommodate passengers. Even transport terminals at CBDs have inadequate facilities and are often characterized as difficult, uncomfortable, and dangerous for passenger movements. Insufficient layover spaces for buses, jeepneys,

and taxis contribute to road congestion as transit vehicles take up road space due to indiscriminate boarding and alighting. These issues reflect the need for an intermodal terminal that would strengthen the link in the intermodal passenger transport chain.

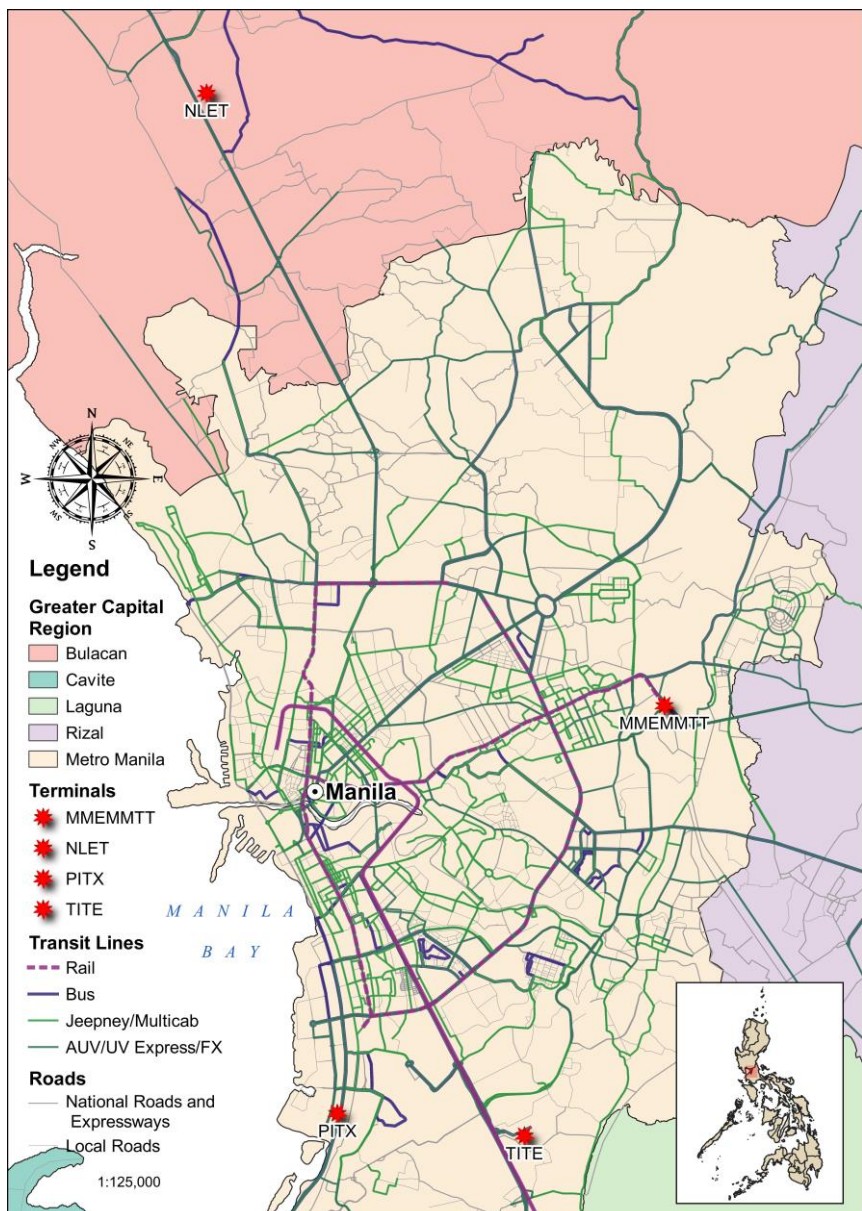

**Figure 2.** Spatial distribution of existing and proposed transport terminals in Mega Manila.

At the time of writing, there are four proposed terminals around the periphery of Metro Manila, as shown in Figure 2. The Parañaque Integration Terminal Exchange (PITX) is an intermodal terminal targeting commuters traveling between Metro Manila and provinces south of the metropolis (Region 4-A). The PITX was also designed to be connected to the planned LRT 1 South extension [56]. The Metro Manila Council also announced the closure of bus terminals along major arterial roads such as the Epifanio de los Santos Avenue (EDSA). Once removed, all buses servicing the regions north of Metro Manila will be relocated to the North Luzon Express Terminal (NLET). Aside from these two terminals, two other terminals were proposed to integrate transport services: the Taguig Integrated Terminal Exchange (TITE) in Food Terminal Inc., Taguig, and the Metro Manila Eastern Multi-Modal Transport Terminal (EMMTT) in Marikina City. Among the four, the PITX is the only multimodal terminal that is operational. Within a few days of its

operations, authorities were bombarded with complaints from provincial bus operators because their route ends did not pass through the terminal. Passengers were also affected as they were suddenly rerouted, and they waited for several hours because of the lack of available buses and jeepneys at the terminals [57]. These issues undoubtedly call into question the effectiveness of the passenger terminal in integrating the various modes of transport and its convenience to passengers.

## 3. Indices for Terminal Evaluation

Using normalized values of the transit metrics, the proximity indices are hereby proposed to be calculated as follows

$$p_{x,i,j} = \sum_{h=1}^{m} \frac{\frac{x_{h,j}}{\sum_{h=1}^{m} x_{h,j}}}{r_i} \tag{1}$$

where $p_{V,i,j}$ is the proximity index for metric $j$ of variable $x$ to terminal $i$, $x_{h,j}$ is the value of metric $j$ for transit stop $h$, $r_i$ is the radial distance between transit stop $h$ and terminal $i$, and $m$ is the total number of transit stops.

The proximity index of variable $x$ to terminal $i$, $p_{V,i}$, is then proposed to be calculated as a summation of the proximity indices of the combined $V$ for all $k$ number of metrics.

$$p_{x,i} = \sum_{h=1}^{m} \frac{\frac{\sum_{j=1}^{k} x_{h,j}}{\sum_{h=1}^{m} \sum_{j=1}^{k} x_{h,j}}}{r_i} \tag{2}$$

In this study, the authors used transit demand ($d$) and transit supply ($s$) as variables. Various passenger activities—namely, (1) initial boarding, (2) transfers, (3) pass through, and (4) final alighting—were used as transit demand metrics. For the transit supply, available transit service capacities for various modes—namely, (1) jeepney, (2) bus, (3) AUV, and (4) train—were proposed to be quantified as follows

$$s_{h,j} = f * C * l \tag{3}$$

where $s_{h,j}$ is the transit supply for metric $j$ at transit stop $h$, computed as the product of the trip frequency ($f$), total capacity ($C$), and transit line length ($l$) available at transit stop $h$.

Proximity indices for each metric $j$ of variable $x$ can also be calculated as follows

$$p_{x,j} = \frac{\sum_{i=1}^{n} \left( \frac{p_{x,i,j}}{\sum_{j=1}^{k} p_{x,i,j}} \right)}{n} \tag{4}$$

where $p_{V,j}$ is the proximity index for metric $j$ of variable $V$ for all $k$ number of metrics. These proposed to be calculated as the average of the proximity index values for each metric for all terminals. Finally, the effective proximity indices for the different metrics can be calculated as follows

$$p_{x,j}^* = p_{x,j} * \frac{\sum_{i=1}^{n} \frac{p_{x,i}}{\sum_{x}^{y} p_{x,i}}}{n} \tag{5}$$

where $n$ is the total number of terminals, $y$ is the total number of variables, and the sum of $\sum_{x}^{y} \sum_{j=1}^{k} p_{x,j}^*$ should be equal to one.

These proximity index values can be used to identify which of the different metrics most closely relate to the spatial distribution of the transport terminals. It involves a simple methodology provided that relevant data (i.e., the spatial distribution of any metric to be analyzed) are available. The procedure quantifies the nearness of any subject to any metric of interest. The proximity index, therefore, could be used to explore the nearness of transport facilities to any metric. This approach may be used alongside any spatial data

gathered through various practices and research methodologies (e.g., sensor-based big data [58–60], remote sensing imagery [61–63]).

## 4. Proximity Index: Evaluating Transit Demand and Supply

For this paper, current passenger activity and transit service operations were used to estimate transit demand and supply metrics, respectively. Passenger activity was classified into four different passenger movements across the road network (e.g., initial boarding, transfer, pass through, and final alighting). These were tallied at each transit stop using the transport modeling software Emme 4.0. Transit services were classified into four categories as well (e.g., jeepney, bus, AUV, and train). More precisely, transit supply was taken as the total transit service capacity available at each transit stop offered by each transit service. Operating characteristics such as headways, capacities, and route lengths were also exported from the Emme model developed.

The Mega Manila transport network was modeled using road alignment data from OpenStreetMap.org, zoning and origin–destination matrices from the Metro Manila Urban Transportation Integration Study Update and Capacity Enhancement Project [53], and transit route and operating characteristics from the Land Transport Franchising and Regulatory Board. The model is composed of 453 centroids, 6950 nodes, and 17,222 links, with 902 transit lines, operated by 11 transit modes. The model was calibrated using volume count, passenger load, and collected onboard survey data. Standard transit assignment was performed to model the spread of the transit demand across the network. All relevant information was then exported to spreadsheet software for further calculations and analyses. A total of 4145 nodes were exported as transit stops (i.e., nodes where at least one transit line passed through and stopped). Transit demand and supply metrics at each transit stop were recorded with the respective x and y coordinates. Figure 3 shows the spatial distribution of the transit demand and supply metrics defined in this study, respectively.

Table 2 shows the proximity indices for each metric and terminal for transit demand, while Table 3 shows those for transit supply, calculated using Equations (1)–(3). As shown, the NLET has significantly low values for both transit demand and supply metrics, which may indicate that it is located far away from the demand it should serve, as well as the supply that could provide the means for it to do so. Also shown are the proximity indices for the transit demand and supply metrics, calculated using Equation (4), with reduced $p_{x,j}$ values when those for the NLET are included. Based on these, the NLET is deemed unfit for use as a terminal location. As such, the authors decided to remove the proposed terminal from subsequent calculations and analyses, as it would unnecessarily distort the resulting values. Only values from the PITX, the TITE, and the EMMT were used for the effective proximity indices for the transit demand and supply metrics, calculated using Equation (5).

**Table 2.** Proximity index of transit demand metrics (baseline).

| Terminal | Transit Demand Metric Proximity Index, $p_{d,i,j}$ | | | | $p_{d,i}$ |
|---|---|---|---|---|---|
| | Initial Boarding | Transfer | Pass Through | Final Alighting | Total |
| PITX | 0.049 | 0.080 | 0.054 | 0.049 | 0.057 |
| TITE | 0.025 | 0.037 | 0.029 | 0.025 | 0.030 |
| EMMTT | 0.086 | 0.109 | 0.125 | 0.087 | 0.122 |
| NLET | 0.011 | 0.004 | 0.006 | 0.010 | 0.006 |
| $p_{d,j}$ (S.D.) | 0.243 (0.062) | 0.267 (0.092) | 0.248 (0.044) | 0.241 (0.055) | 0.489 (0.140) |
| $p_{d,j}$ w/o NLET (S.D.) | 0.213 (0.003) | 0.308 (0.042) | 0.262 (0.035) | 0.215 (0.003) | 0.432 (0.100) |
| $p_{s,j}^{*}$ | 0.092 | 0.133 | 0.114 | 0.093 | |

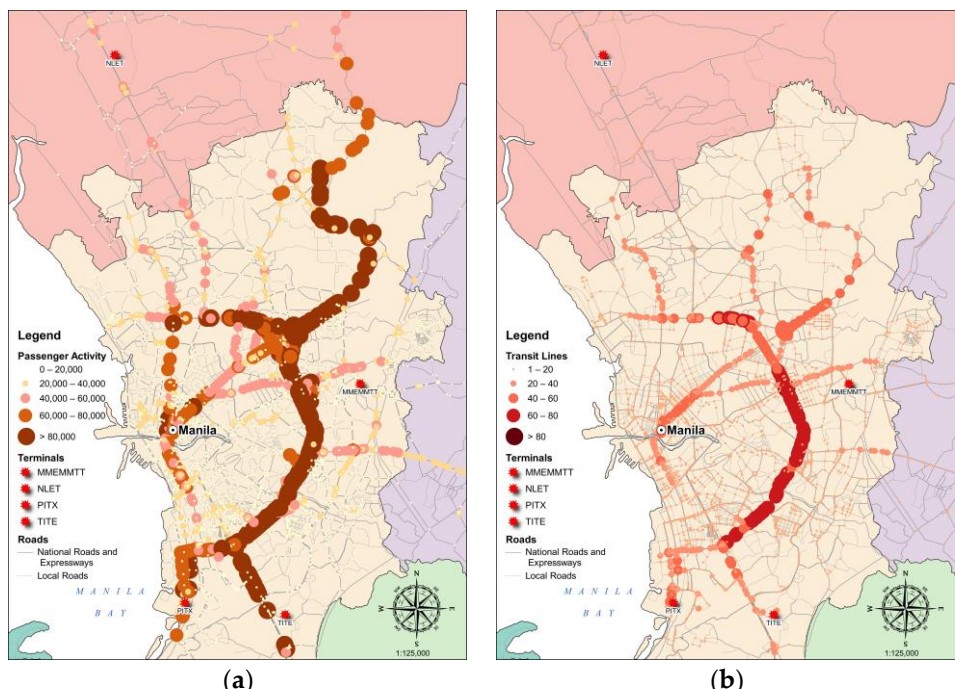

**Figure 3.** Spatial distribution of transit demand and supply in the study area. (**a**) Public transport passenger activity and (**b**) public transport services.

From these values, the spatial distribution of the transport terminals in the study area is more closely related to the transit supply (i.e., 0.567 versus 0.432). The difference between the two values, on the other hand, can be interpreted as the degree of the mismatch between the transit supply and demand. This may indicate the transport system's inability to meet the appropriate passenger demand, which may be interpreted as the transport planners' failure to anticipate travel demand growth in areas not readily served by the existing transit services.

Looking at the effective proximity indices for the transit demand metrics, the spatial distribution of the transport terminals is least related to both where trips begin and end. This shows that the locations of these terminals are not readily accessible where travelers initially come from and/or eventually go. This may reflect instances wherein the terminals are ineffective as a transport hub due to the relatively low access to these terminals.

**Table 3.** Proximity index of transit supply metrics (baseline).

| Terminal | Transit Supply Metric Proximity Index, $p_{s,i,j}$ | | | | $p_{s,i}$ |
|---|---|---|---|---|---|
| | **Jeepney** | **Bus** | **AUV** | **Train** | **Total** |
| PITX | 0.058 | 0.074 | 0.078 | 0.063 | 0.073 |
| TITE | 0.079 | 0.061 | 0.062 | 0.024 | 0.061 |
| EMMTT | 0.117 | 0.108 | 0.108 | 0.136 | 0.109 |
| NLET | 0.003 | 0.003 | 0.009 | 0.000 | 0.003 |
| $p_{s,j}$ (S.D.) | 0.250 (0.071) | 0.244 (0.030) | 0.347 (0.171) | 0.157 (0.129) | 0.510 (0.140) |
| $p_{s,j}$ w/o NLET (S.D.) | 0.270 (0.071) | 0.256 (0.022) | 0.263 (0.026) | 0.209 (0.091) | 0.567 (0.100) |
| $p_{s,j}^*$ | 0.154 | 0.146 | 0.150 | 0.119 | |

As for the effective proximity indices for the transit supply metrics, the highest relation can be identified to that of the jeepney service, followed by that of the AUV service. The

bus service follows while that of train lags falls far behind. This can be interpreted as the poor integration of the terminal location with the mass transport systems available in the study area. Despite having the highest passenger service share, the bus service falls behind in comparison with the jeepney and AUV in its relation to the locations of the terminals. Presumably, the proposed location of the terminal will not properly integrate mass transit options such as rail and bus services into the rest of the system. The locations, therefore, favor services with lower passenger capacities. With bus services ranking behind jeepney and AUV counterparts, patronage to the former services could be anticipated to decline.

## 5. Proposed Location of New Transport Terminals

Using these metrics, this paper proposes locations of new transport terminals (see Figure 4) to be better integrated with the public transport system in the study area. Firstly, the NLET is notably far from the metropolitan area; it is quite distant from both transit demand and supply. As such, the authors recommend the development of another transport terminal to serve the traffic demands coming from and/or going to areas north of Metro Manila. Situated right at the outskirt of the region, Balintawak, Quezon City is an area of significant passenger activity, providing access to different transport modes traversing across Metro Manila. Within the vicinity, train and bus lines are more easily accessible.

Shown in Tables 4 and 5 are the proximity index values for the Balintawak terminal (BLT), as well as the new effective proximity indices for the whole study area. The $p_{x,i}$ values for the proposed terminal were found to be the highest for those of both transit demand and supply metrics, which indicates that it is even better located than the rest (i.e., nearer to transit demand and supply). Additionally, it can be seen that the addition of the BLT will more closely relate the spatial distribution of transport terminals to the transit supply (i.e., 0.574 versus 0.425), with a significant increase in $p_{d,j}$ values, specifically for the bus and train lines (i.e., increasing from 0.145 to 0.158 and from 0.119 to 0.132, respectively). This means that it will provide better integration with the mass transit systems operating in the study area. Additionally, considering the values for transit demand metrics, a terminal in the vicinity improves its relationship with initial boarding and final alighting of travelers (i.e., increasing from 0.092 to 0.094 and from 0.092 to 0.095, respectively). This terminal location can be interpreted as better integrated with where trips start and/or end. With all these considerations, the authors propose the BLT as a viable replacement for the proposed NLET.

**Table 4.** Proximity index of transit demand metrics (with the Balintawak terminal (BLT)).

| Terminal | Transit Supply Metric Proximity Index, $p_{d,i,j}$ | | | | $p_{d,i}$ |
|---|---|---|---|---|---|
| | Initial Boarding | Transfer | Pass Through | Final Alighting | Total |
| PITX | 0.041 | 0.071 | 0.051 | 0.042 | 0.053 |
| TITE | 0.025 | 0.037 | 0.029 | 0.025 | 0.030 |
| EMMTT | 0.057 | 0.078 | 0.097 | 0.057 | 0.093 |
| BLT | 0.155 | 0.135 | 0.119 | 0.157 | 0.123 |
| $p_{d,j}$ (S.D.) | 0.221 (0.035) | 0.293 (0.047) | 0.261 (0.052) | 0.223 (0.036) | 0.425 (0.087) |
| $p_{d,j}^*$ | 0.094 | 0.125 | 0.111 | 0.095 | |

However, looking at the proximity indices for transit demand and supply in general, this location veers toward the transit supply instead of the demand. As such, the authors propose another terminal right at the heart of the region. Located in Paco, Manila, the old Paco Railway station has been dormant since 1996. With an area of approximately 2 hectares, this patch of government-owned land can be used to house a central terminal, such as those in Seoul, South Korea, and Tokyo, among many others. Furthermore, being right across the Plaza Dilao exit of the Metro Manila Skyway Stage 3, connecting the North

Luzon Expressway and South Luzon Expressway, this terminal can serve as pick-up or drop-off points even for provincial trips.

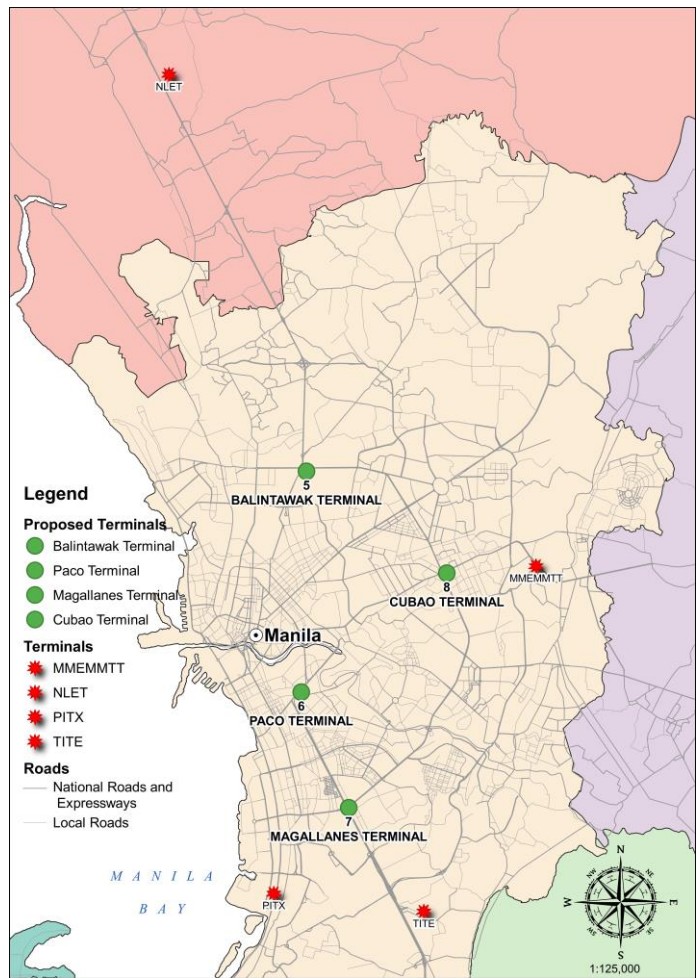

**Figure 4.** Spatial distribution of proposed new transport terminals.

**Table 5.** Proximity index of transit supply metrics (with the BLT).

| Terminal | Transit Supply Metric Proximity Index, $p_{s,i,j}$ | | | | $p_{s,i}$ |
|---|---|---|---|---|---|
| | **Jeepney** | **Bus** | **AUV** | **Train** | **Total** |
| PITX | 0.050 | 0.073 | 0.076 | 0.052 | 0.071 |
| TITE | 0.079 | 0.061 | 0.062 | 0.024 | 0.061 |
| EMMTT | 0.077 | 0.078 | 0.084 | 0.102 | 0.079 |
| BLT | 0.111 | 0.183 | 0.112 | 0.177 | 0.179 |
| $p_{s,j}$ (S.D.) | 0.241 (0.073) | 0.275 (0.035) | 0.254 (0.047) | 0.229 (0.092) | 0.574 (0.087) |
| $p_{s,j}^*$ | 0.139 | 0.158 | 0.146 | 0.132 | |

Tables 6 and 7 show the updated proximity index values, including those for the Paco terminal (PCT). Though this proposed terminal only has the second-highest $p_{x,i}$ values, its addition still has merits, based on the effect on the overall ratings of the different metrics, particularly the shift towards transport demand (i.e., increasing from 0.432 to 0.448). This shows that putting up a central terminal more closely relates to the actual transit demand that it is supposed to serve in the first place. Moreover, aside from better integration with the trip ends as shown in the increase in proximity indices for initial boarding and final

alighting (i.e., increasing from 0.092 to 0.100 and from 0.093 to 0.100, respectively), this location can also serve as a hub for existing transfers occurring within the metropolitan area (i.e., increasing from 0.133 to 0.135). It also has the potential to make a significant contribution to the reintegration of the existing railways, as shown in the proximity value for transit demand for trains (i.e., increasing from 0.119 to 0.140).

**Table 6.** Proximity index of transit demand metrics (with the Paco terminal (PCT)).

| Terminal | Transit Supply Metric Proximity Index, $p_{d,i,j}$ | | | | $p_{d,i}$ |
|---|---|---|---|---|---|
| | Initial Boarding | Transfer | Pass Through | Final Alighting | Total |
| PITX | 0.032 | 0.062 | 0.041 | 0.032 | 0.043 |
| TITE | 0.020 | 0.025 | 0.019 | 0.020 | 0.020 |
| EMMTT | 0.069 | 0.095 | 0.110 | 0.069 | 0.106 |
| PCT | 0.108 | 0.104 | 0.087 | 0.108 | 0.090 |
| $p_{d,j}$ (S.D.) | 0.222 (0.034) | 0.301 (0.050) | 0.251 (0.047) | 0.223 (0.034) | 0.448 (0.114) |
| $p_{d,j}^*$ | 0.100 | 0.135 | 0.113 | 0.100 | |

**Table 7.** Proximity index of transit supply metrics (with the PCT).

| Terminal | Transit Supply Metric Proximity Index, $p_{s,i,j}$ | | | | $p_{s,i}$ |
|---|---|---|---|---|---|
| | Jeepney | Bus | AUV | Train | Total |
| PITX | 0.041 | 0.064 | 0.073 | 0.017 | 0.062 |
| TITE | 0.078 | 0.044 | 0.050 | 0.024 | 0.045 |
| EMMTT | 0.091 | 0.094 | 0.105 | 0.101 | 0.094 |
| PCT | 0.121 | 0.067 | 0.039 | 0.274 | 0.074 |
| $p_{s,j}$ (S.D.) | 0.269 (0.084) | 0.232 (0.080) | 0.244 (0.123) | 0.253 (0.208) | 0.551 (0.114) |
| $p_{s,j}^*$ | 0.149 | 0.128 | 0.135 | 0.140 | |

Another possible location for a transport terminal within the metropolitan area is at the Magallanes Interchange in Makati City, located at the intersection of the EDSA and Osmena Highway. Within its vicinity are access points to two of the existing four train lines, as well as the majority of the bus routes plying the EDSA. This area is also close to the Makati CBD where a significant chunk of economic activities in the whole region transpires throughout the day.

Tables 8 and 9 show the new proximity index values for all terminals, including those for the Magallanes terminal (MGT), where a significant shift towards transport supply can be seen. This may indicate an increase in the transit coverage that can be served by the terminal. However, the reduction in values for transit demand metrics, particularly for the initial boarding and final alighting, shows that the MGT does not necessarily serve as a catchment for trips, but rather as a transfer station for in-between trips. As shown in the $p_{x,i}$ values, the proposed terminal rates are higher than the other three, showing that it has better proximity to both the transit supply and demand than the other terminals. It is also worth mentioning that the MGT's inclusion resulted in a reduction in values of that of the PITX and the TITE. This may indicate that, in a scenario where it is possible to replace currently existing infrastructures with one that can serve its purpose more effectively, the MGT fares better than having both the PITX and TITE combined. However, with the PITX already in operation, the authors propose the MGT as a replacement for the proposed TITE. Looking at the proximity index values for transit supply metrics, there was effectively a reduction in that of trains, despite being accessible to two train lines. This can be explained

as the transit supply being offered by the bus and AUV lines passing through the area greatly surpassing that offered by the train. This indicates that there may be too many bus and AUV routes.

**Table 8.** Proximity index of transit demand metrics (with the Magallanes terminal (MGT)).

| Terminal | Transit Demand Metric Proximity Index, $p_{d,i,j}$ | | | | $p_{d,i}$ |
|---|---|---|---|---|---|
| | Initial Boarding | Transfer | Pass Through | Final Alighting | Total |
| PITX | 0.019 | 0.050 | 0.031 | 0.019 | 0.032 |
| TITE | 0.018 | 0.020 | 0.014 | 0.018 | 0.015 |
| EMMTT | 0.080 | 0.099 | 0.118 | 0.080 | 0.114 |
| MGT | 0.115 | 0.214 | 0.146 | 0.116 | 0.153 |
| $p_{d,j}$ (S.D.) | 0.205 (0.038) | 0.333 (0.070) | 0.253 (0.046) | 0.206 (0.039) | 0.406 (0.097) |
| $p_{d,j}^*$ | 0.084 | 0.136 | 0.103 | 0.084 | |

**Table 9.** Proximity index of transit supply metrics (with MGT).

| Terminal | Transit Supply Metric Proximity Index, $p_{s,i,j}$ | | | | $p_{s,i}$ |
|---|---|---|---|---|---|
| | Jeepney | Bus | AUV | Train | Total |
| PITX | 0.014 | 0.048 | 0.065 | 0.000 | 0.045 |
| TITE | 0.067 | 0.035 | 0.043 | 0.021 | 0.036 |
| EMMTT | 0.103 | 0.103 | 0.108 | 0.116 | 0.103 |
| MGT | 0.188 | 0.236 | 0.160 | 0.166 | 0.232 |
| $p_{s,j}$ (S.D.) | 0.250 (0.119) | 0.285 (0.074) | 0.308 (0.137) | 0.155 (0.117) | 0.593 (0.097) |
| $p_{s,j}^*$ | 0.148 | 0.170 | 0.183 | 0.092 | |

Like the MGT, a terminal in Cubao, Quezon City will be in a prime position to cater to two of the existing four train lines, as well as a greater majority of the bus lines plying the EDSA. Located at the intersection of the EDSA and Aurora Blvd., this location is also accessible by jeepney and AUV transit lines plying east to west and vice versa. This may also serve as a gateway for long-haul trips to the east of the region. This location's ultimate drawback, however, may be that there is no available space anywhere in its vicinity where the terminal can be constructed. There are currently no open spaces that may be easily occupied. Ergo, a terminal in this location may only be feasible in partnership with the private institutions owning the land, where the lower floors of a large-scale development can be allotted for public use.

Tables 10 and 11 show the updated proximity index values, where it can be seen how the Cubao terminal (CBT) rates higher than all other three terminals combined. This shows how extremely suitable this area is to serve both the transit demand and supply in the region. In addition, this location is where the transit demand is most concentrated, as shown in the significant change in transit demand metric (i.e., increasing from 0.432 to 0.477). This 0.477–0.522 relationship between transit demand and supply is nearest to the ideal 0.5–0.5 among all terminal configurations considered in this study. Moreover, a significant reduction in index $p_{x,i}$ values for the EMMTT may indicate that the CBT is in a better location to cater to the prospective demand for the EMMTT. Thus, the authors recommend the CBT as a viable replacement for the proposed EMMTT. Looking at the proximity index values individually, those for passenger transfer rate significantly higher than others in transit demand, which affirms this location as a viable transit hub. As for the transit supply metrics, an almost even reduction among all modes can be seen.

**Table 10.** Proximity index of transit demand metrics (with the Cubao terminal (CBT)).

| Terminal | Transit Supply Metric Proximity Index, $p_{d,i,j}$ | | | | $p_{d,i}$ |
|---|---|---|---|---|---|
| | Initial Boarding | Transfer | Pass Through | Final Alighting | Total |
| PITX | 0.041 | 0.071 | 0.051 | 0.041 | 0.053 |
| TITE | 0.023 | 0.035 | 0.027 | 0.023 | 0.028 |
| EMMTT | 0.027 | 0.036 | 0.039 | 0.027 | 0.038 |
| CBT | 0.135 | 0.453 | 0.386 | 0.137 | 0.382 |
| $p_{d,j}$ (S.D.) | 0.187 (0.044) | 0.339 (0.054) | 0.287 (0.047) | 0.186 (0.042) | 0.477 (0.175) |
| $p_{d,j}^{*}$ | 0.089 | 0.162 | 0.137 | 0.089 | |

**Table 11.** Proximity index of transit supply metrics (with the CBT).

| Terminal | Transit Supply Metric Proximity Index, $p_{s,i,j}$ | | | | $p_{s,i}$ |
|---|---|---|---|---|---|
| | Jeepney | Bus | AUV | Train | Total |
| PITX | 0.050 | 0.073 | 0.076 | 0.051 | 0.071 |
| TITE | 0.078 | 0.058 | 0.061 | 0.024 | 0.058 |
| EMMTT | 0.050 | 0.011 | 0.065 | 0.076 | 0.014 |
| CBT | 0.203 | 0.524 | 0.113 | 0.210 | 0.499 |
| $p_{s,j}$ (S.D.) | 0.248 (0.073) | 0.275 (0.182) | 0.253 (0.098) | 0.222 (0.112) | 0.522 (0.175) |
| $p_{s,j}^{*}$ | 0.130 | 0.144 | 0.132 | 0.116 | |

## 6. Conclusions and Recommendations

The proposed method evaluates the proximity of the public transport terminals to existing routes and passenger movements at different points in the transport network. The resulting values can be used to interpret which metrics (e.g., transit demand, transit supply) are relatively more closely related to the spatial distribution of the transit terminals. Using these values as guidance measures, transit system deficiencies (i.e., in terms of its inability to cater to either the demand or supply) can be identified and addressed.

Specific to this study, the BLT better integrates into the mass transit systems and where trips start and/or end. The PCT serves as a hub for existing transfers occurring within the metropolitan area while potentially contributing significantly to the reintegration of the existing railways. The MGT is effective as a transfer station connecting bus, AUV, and train lines. Lastly, the CBT's location right in the middle of the an area with high transit demand makes it a very viable transit hub. With this study's results, the authors reiterate the viability of the BLT, MGT, and CBT, as these terminal locations were found to be better fit to the existing transit demand and supply than the NLET in the North, the TITE in the South-East, and the EMMTT in the East, respectively.

Alternatively, should the planned terminals be pursued, the authors suggest designing additional transit routes that would pass through the terminal or realignment of existing transit routes to extend into the terminal. Development of surrounding areas to increase transit demand within the immediate vicinity of the terminal using the TOD concept could also be considered to balance out the existing mismatch. Addressing the imbalance makes for a more efficient and sustainable design of a public transport system.

Finally, this methodology quantifies the nearness of any subject to any metric of interest. This can be employed to explore spatial options using proximity index values as indicators of nearness to any preferred metric. For example, a person can use this approach to compare different properties by quantifying the terminal's attractiveness based on its proximity to any facility and/or activity of interest (e.g., market, school, etc.).

This methodology can also be applied using various other transport infrastructure metrics such as road capacities (i.e., to encourage possible modal shifting from private car use) or levels-of-service/speeds (i.e., to prioritize improvement of highly congested highways).

Further research may also be conducted on qualifying these indices to determine the optimal values (0.5–0.5) by applying proximity indexing on transit networks widely accepted to be effective and efficient. Multi-criteria analyses or any other similar research methodologies may also be employed in determining optimal index values should three or more metrics be analyzed.

**Author Contributions:** Conceptualization, K.I.D.R. and R.P.A.; Data curation, K.I.D.R.; Formal analysis, K.I.D.R.; Funding acquisition, A.F.; Investigation, K.I.D.R. and R.P.A.; Methodology, K.I.D.R.; Project administration, A.F.; Supervision, A.F.; Validation, R.P.A.; Visualization, R.P.A.; Writing—original draft, K.I.D.R.; Writing—review & editing, K.I.D.R., R.P.A. and A.F. All authors have read and agreed to the published version of the manuscript.

**Funding:** This research has received no external funding.

**Institutional Review Board Statement:** Not applicable.

**Informed Consent Statement:** Not applicable.

**Acknowledgments:** The authors extend their gratitude to DOST-PCIEERD for funding the Metro Manila Transportation Network—Big Data Analytics and Application (MMTN-BDAA) Project and the De La Salle University Science Foundation Inc. for the publication support.

**Conflicts of Interest:** The authors declare no conflict of interest.

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
