# Peer review of "Proximity Indexing of Public Transport Terminals in Metro Manila"

_sustainability, doi:10.3390/su13084216_

Round 1

Reviewer 1 Report

Congratulations on a well-documented, well-written, well-prepared and well-explained paper.  Your methodology can and will be used by many municipalities.

Reviewer 2 Report

In this work, a method proposed that evaluates the location of the terminal based on its proximity to both transit supply (public transport services) and demand (public transport passenger activity). Based on their founding, the spatial distribution of the transport terminals in the study area is more closely related to the transit supply. The authors proposed an interesting method, however, some information is missing and a revision is required. Please consider my comments and edit your manuscript according to my comments to be ready for publication. In the abstract, please discuss more the methodology which used in this work, and what are the differences with other methods. In the introduction, there are many references mentioned without discussing their methodology, please discuss them briefly. Also, please include some other references about aerial traffic analysis, for this purpose, please include the following papers: Incident detection algorithm based on radon transform using high-resolution remote sensing imagery, Incident and traffic-bottleneck detection algorithm in high-resolution remote sensing imagery, and, Road-following and traffic analysis using high-resolution remote sensing imagery. Please include the reference/source of all Figures in their captions. Please include reference(s) for the Equations if applicable. You may combine some Equations (numbers). 

Reviewer 3 Report

This article is read and structured well, and the proposed sites were examined which added values to the research.

I only have few comments as follows.

  1. From the article I knew that Terminal 4 is the NLET, but how about Terminal 1, 2 and 3?
  2. It seems that four proposed sites were replacing the existing Terminal 4 in the simulation, what happened to Terminal 4 after the replacement? Was Terminal 4 downgraded as a normal station or something else?  What would happen if Terminal 4 had been in the simulation?
  3. The relationship between Groups A (Terminal 1, 2 and 3) and Group B (Terminal 5, 6, 7 and 8) was not in the discussion. What I meant is, as soon as Group B is in the simulation, to what extent are Terminal 1, 2 and 3 required to adjust to the new relationship?  Do Terminal 1, 2 and 3 need to find new proposed terminals?  Whether the existence of Terminal 5, 6, 7 and 8 change the role of Terminal 1, 2 and 3 in their surrounding areas would be an interesting issue.4
  4. The conclusion and recommendation section was a little bit vague and could be more focused on the empirical results.

Round 2

Reviewer 2 Report

Some of my comments have not addressed properly. Please revise your manuscript according to the comments. Please expand Figure's captions and include more information. Some text in Figures hardly readable. If there is more than one image, number them accordingly and discuss them in the caption. 
